# “Our Work, Our Health, No One’s Concern”: Domestic Waste Collectors’ Perceptions of Occupational Safety and Self-Reported Health Issues in an Urban Town in Ghana

**DOI:** 10.3390/ijerph19116539

**Published:** 2022-05-27

**Authors:** Samuel Yaw Lissah, Martin Amogre Ayanore, John K. Krugu, Matilda Aberese-Ako, Robert A. C. Ruiter

**Affiliations:** 1Department of Work and Social Psychology, Faculty of Psychology and Neuroscience, Maastricht University, P.O. Box 616, 6200 MD Maastricht, The Netherlands; r.ruiter@maastrichtuniversity.nl; 2Department of Agro-Enterprise Development, Faculty of Applied Sciences and Technology, Ho Technical University, P.O. Box HP 217 Ho, Volta Region, Ghana; 3Department of Health Policy Planning and Management, School of Public Health, University of Health and Allied Sciences, PMB 31 Ho, Volta Region, Ghana; mayanore@uhas.edu.gh; 4Centre for Health Policy Advocacy Innovation & Research in Africa (CHPAIR-Africa), 441/4F Nyaniba Estates, Osu, Accra, Greater Accra Region, Ghana; 5KIT Royal Tropical Institute, P.O. Box 95001, 1090 HA Amsterdam, The Netherlands; j.krugu@kit.nl; 6Institute of Health Research, University of Health and Allied Sciences, PMB 31 Ho, Volta Region, Ghana; maberese-ako@uhas.edu.gh

**Keywords:** domestic waste collectors, occupational health and safety, self-reported health, qualitative methods, Ghana

## Abstract

Domestic waste collectors face major public health hazards that result in injuries and morbidity globally. This study explored domestic waste collectors’ perceptions of occupational safety and self-reported health issues in a city in Ghana using a phenomenological qualitative research design. In-depth interviews and focus group discussions were held with 64 domestic waste collectors from two waste companies. The transcribed data were imported into NVivo 11.0 software (QSR International, Burlington, MA, USA) for coding, and a content analysis was applied to analyze all the transcribed data using the processes of induction and deduction. The consensual views from the domestic waste collectors showed the waste-company employers’ non-interest in the domestic waste collectors’ occupational safety and health. Poor communication from employers to domestic waste collectors and huge workloads were identified as the causes of the poor implementation of occupational safety practices, which exposed the domestic waste collectors to occupational health hazards. The domestic waste collectors reported that they suffered from occupational injuries, psychosocial disorders, work-related stress, and frequent burnout. The domestic waste collectors adopted coping strategies, such as self-medication, to deal with these occupational hazards, since most of them were not covered by guaranteed health insurance. In addition, the study revealed the non-compliance and non-enforcement of occupational health and safety policies by the employers to guide health and safety training and practices among the domestic waste collectors. In conclusion, the findings suggest that DWCs are exposed to occupational safety and health hazards in their work. Waste-company employers should extend welfare benefits to DWCs, such as health insurance and social security benefits, to ensure their security, health, and well-being. The findings could inform the design of intervention programs and policies to guide training and practices for domestic waste collectors.

## 1. Introduction

Occupational injuries, fatigue, accidents, and disease morbidity outcomes are major public health problems that waste collectors face globally [1,2,3]. Unfortunately, evidence suggests that domestic waste collectors (DWCs) are less protected against occupational health risks [4,5,6], directly or indirectly, during the collection and processing of waste to final dumping sites [7]. Rushton [8] estimates that, annually, about 270 million waste collectors worldwide suffer from occupational injuries, accidents, and diseases, with about 2.3 million estimated to die annually from the hazardous effects of the job.

Compared to other employees in other work environments, waste collectors disproportionately suffer from health hazards and injuries from work-related accidents. For example, occupational accidents and injuries among DWCs were estimated to be 5.6 times higher compared to those of other workers in a study conducted in Denmark [9]. A study conducted in Zimbabwe in the informal sector also reported that garbage (waste) workers experience a high incidence of repetitive strain injuries due to repeated flexing and twisting motions as a direct consequence of their work [10]. Other common health-related problems reported in other studies among DWCs include musculoskeletal complaints, asthma, gastrointestinal problems, eye infections, skin irritation, fever, cough, and fatigue [11,12,13,14]. In their study, Goldstein et al. [15] found that occupational diseases, accidents, and injuries among DWCs are high due to neglect of the occupational health needs among this group of workers by employers and the regulatory institutions responsible for enforcing safety standards. The health-related risks among DWCs are influenced by socio-cultural factors, such as norms and beliefs that contribute to negative attitudes toward solid-waste management [15].

Managing the waste generated in households or public space in Ghana is challenging due to socio-cultural, institutional, and legislative bottlenecks [16,17]. For example, studies in Ghana found the weak enforcement of sanitation laws, the poor attitudes of citizens toward generating, managing, and disposing of waste, and the labor-intensive nature of waste collection as some of the reasons why solid-waste management is still a problem [7,18,19,20,21,22]. Inadequate funding, non-availability, and poor infrastructure for waste collection and processing negatively impact the safe collection and transportation of solid waste, and the infrastructure of waste collection and processing negatively impacts the safe collection and transportation of solid waste [21,22,23,24]. While dealing efficiently and effectively with urban solid waste promotes human and environmental security [25,26] and advances health and well-being, the occupational and health risks posed by poor waste management practices and their effects on DWCs are understudied in Ghana.

Despite the significant amount of evidence on the health hazards faced by workers in the sanitation industry, there is a paucity of evidence examining the self-reported health and occupational safety aspects among DWCs in the Ghanaian context. One study found an association between waste burning and the incidence of respiratory health symptoms among DWCs [27]. Lissah et al. [28] reported on DWCs’ experiences of work-related stress and heightened psychosocial risk. The awareness and attitudes of the population on waste-related health risks have been reported in Ghana [29].

The literature in Ghana on the self-reported health risks among waste collectors is limited to Kretchy et al.’s study [30] on solid-waste handlers’ self-reported health issues. Their findings suggest that waste handlers need affordable and suitable protective gear to reduce the risk to their health, as well as the provision of water and soap to promote personal hygiene at work. Kretchy et al. did not explore the perceptions of domestic waste collectors regarding work safety and health challenges in the work environment. Knowledge of the perceptions, beliefs, and behaviors of DWCs can inform policy recommendations on how employers can support and promote healthy work environments among waste workers in Ghana. DWCs are exposed to occupational safety and health hazards in their work due to the huge volumes of solid waste resulting from the ever-growing population [31,32]. The significant increase in the quantity of solid waste has exerted a profound effect on occupational health and safety issues in the solid waste collection and management sector. Several studies have been developed with regard to the potential health hazards of the improper disposal of solid waste for the environment and the public. However, the occupational safety and health hazards associated with solid-waste collection jobs have received little attention. Therefore, this qualitative study aimed to examine and understand DWCs’ perceptions of occupational safety and self-reported health outcomes in a city in Ghana. Our findings add to the literature on occupational hazards in the context of solid-waste management practices in under-resourced countries, such as Ghana. They further provide municipal authorities with some context-based experiences for improving the health and safety needs of DWCs. The findings are also expected to inform policies on promoting a healthy and productive work-based culture that supports the tenets of a sustainable environment and guarantees safe workplaces.

## 2. Materials and Methods

### 2.1. Research Design and Setting

This study employed a phenomenological qualitative research design. Phenomenology as a qualitative inquiry approach focuses on finding commonalities around the lived experiences of participants studied in a population [33]. The use of phenomenology was appropriate because it enabled the research team to elicit information on individual experiences regarding occupational safety and health issues, and how these impact on overall waste-management systems in Ghana by giving a “voice” to participants, allowing them to speak “freely”.

The study was conducted in the Ho Municipality of the Volta Region of Ghana. At present, the Volta Region has 18 administrative districts, with the Ho Municipality as the administrative capital. At the time of data collection, the projected 2018 population of the Ho Municipality stood at 213,960, 51 percent of whom were female [34]. Due to its strategic location and proximity to the Republic of Togo, as well as easy access to district and regional capitals, such as Accra, the municipality is recognized as the commercial hub of the Volta Region. The municipality was chosen for the study because of its solid-waste management issues and the state of the municipality’s environment, both of which require significant improvement. Two waste management companies operate in the Ho Municipality, company A and company B, which also comprised the study sites. Further details of the two companies can be found in a study by Lissah et al. [28].

#### Ethical Consideration

The study obtained ethical approval from the Ghana Health Service Institutional Review Board (IRB) (GHSERC 08/05/17) and the Ethics Review Committee of Psychology and Neuroscience (ERCPN 188_10_02_2018) at Maastricht University, the Netherlands. Each participant provided signed or thumb-printed consent. During the interviews, the participants were taken through informed consent process, which detailed the study’s aims, risks, benefits, and right to decline at any point in time without any prejudice. After acquiring their written consents, those who consented were recruited voluntarily. Confidentiality, privacy, and anonymity (details that could reveal the identity of the study participants were omitted in transcribing the audio-recorded interviews) were guaranteed to study participants at all times.

### 2.2. Population

The study population in this study has been described in more detail by Lissah et al. [28]. DWCs worked in two waste-management companies in the Ho Municipality of Ghana. The two waste-management companies have been operating in the Ho Municipality for the last 12 years. DWCs are the persons employed by the Assemblies or private solid-waste companies to collect and dispose of solid waste (including cleaners, sweepers, janitors, and drivers, who worked for the two waste companies processing solid waste along the value chain). Their daily work schedules involve the collection, processing, transportation, and disposal of waste in designated dumping sites. Furthermore, DWCs’ work involves heavy lifting, pulling, and manual handling of solid waste containers. As a result, DWCs are more likely to be exposed to occupational safety and health hazards. Due to the presence of a high rate of unemployment, widespread poverty, and the lack of a safety net for the poor, DWCs carry out solid-waste collection to ensure their livelihood in third-world countries, such as Ghana, which presents them with long-term exposure to health risks. The demographic profiles of DWCs’ from the two waste-management companies show they are among the least educated and most poorly remunerated workforces [28,35] To be eligible to participate in the study, domestic waste collectors needed to have worked for one year or more in a solid-waste company.

### 2.3. Data Sampling and Collection Procedures

The list of all DWCs’ names was obtained from waste company managers and supervisors after the two companies received detailed information on the study’s aim and objectives and both companies agreed to participate in the study. In total, 64 DWCs were interviewed among the two companies. The study comprised a total of 43 DWCs from company A and 21 from company B. To minimize bias, two types of non-random sampling technique [33,34] were used to select the study participants. Given that the two waste companies had different levels of capacity and infrastructure, quota sampling, also referred to as proportionate sampling, was first used to select a number of study participants in proportion to the DWC population of each company. This was followed by the use of convenience sampling to select waste collectors per company for the study (Table 1). Convenience sampling is a non-probabilistic sampling method that involves recruiting members of the population who are easily available for data collection [33,34]. The use of the two sampling techniques helped to reduce bias in the selection of study participants.

Data were collected from DWCs in the two waste companies operating in the Ho Municipality. Focus group discussions (FGDs) and follow-up in-depth interviews (IDIs) were conducted among eligible DWCs from the two companies using interview guides designed and peer-reviewed by subject experts prior to pilot testing in the field. The instrument was pilot-tested in Ewe and English languages in both companies before field data collection. Except for some minor textual adjustments, there were no changes to the study instruments. A total of 15 DWCs participated in IDIs and 49 participated in FGDs (see Table 1). On average, each IDI lasted 20 min, while FGDs lasted for 60 min. All interviews were further refined until we reached saturation, where no new information emerged during the interview [34,35,36].

To assist with data collection, three research assistants with indigenous knowledge of the setting and fluency in the language were trained. Research assistants were involved in administering consent forms and taking detailed notes during the interviews. The data analyzed in this study were collected on the occupational safety and self-reported health outcomes of DWCs. Table 2 outlines key sample questions that were explored in this study. Data collection was conducted in convenient places within the company premises or at participants’ preferred locations; they were assured of confidentiality.

The key themes that were explored in the interviews were preventive measures for safety and health hazards at the workplace, DWCs’ perceptions of occupational safety, and health issues.

### 2.4. Data Analysis

Two research assistants transcribed all audiotaped interviews. The principal investigator (SYL) validated an initial coding for transcripts to ensure the consistency and validity of transcribed data. A coding scheme was developed to guide the coding for each IDI and FGD interview. All transcripts were then imported into QSR NVivo 11. In NVivo, further induction (open coding) for participant concepts was performed. Constant comparison enabled the responses to be further categorized into themes [21]. We used the model explorer tool to map out how the participants’ views related to each other our study’s aim [36]. To guarantee the reliability of our results, we estimated the interrater coding reliability for all coded data [37]. Results were presented in a thematic form, which were supported with quotes. Further comparison during the analysis of concepts, codes, and nodes into themes in NVivo resulted in three broad themes: “Our work, our health, no one’s concern”, self-reported health issues and occupational safety coping strategies, and the lack of prioritization of occupational health and safety policies (OHSP) and practices in the workplace. The Section 3 presents the themes that emerged from the study in detail. Figure 1 presents a framework of factors in the study that influenced DWCs’ perceptions of occupational safety and self-reported health outcomes and the details of the emergent themes are presented in the subsequent sub-sections.

## 3. Results

### 3.1. Background of Study Participants

Data from the two waste companies show that the majority of participants were married (40/64) and female (42/64). The educational levels of the participants varied, ranging from 16 participants who had middle-school leaving certificates (MSLC), 14 who had completed junior high school (JHS), five had vocational-level training, and 29 participants were never educated. Three participants reported belonging to a different ethnic group, while the majority reported belonging to the Ewe ethnic group (61/64). Lissah et al. [28] previously presented the socio-demographic details of the 64 DWCs described in this section in a study that examined psychosocial risk, work-related stress, and job satisfaction among the same study participants (see Table 3). This study presents findings on occupational safety and self-reported health outcomes in the same group of DWCs.

#### 3.1.1. Emergent Theme 1: “Our Work, Our Health, No One’s Concern”

The 64 study participants reported that their employers did not show any concern for the occupational hazards and consequences that affect their health and well-being. The DWCs’ views were due to their perceived neglect of the health consequences of their routine waste collection and disposal roles, which has reflected the reduced prioritization of their health by their employers. The waste collectors expressed a strong perception that their health was in their hands, which explained their lack of trust or interest in placing their health and well-being in the hands of their employers, given that their employers had no concern for them.


*“As waste collectors, our concern is our health. For them (employers), our work, our health, is not their concern. All that they are interested in is the streets are clean every morning. Nothing else.”*
(Female FGD participant from company B)

The waste collectors identified that their health was primarily their responsibility, with little or no support from their employer. A quote from a domestic waste collector arguably explains their conviction that no one considered their health:


*“Our work, our health, no one’s concern, but for us (waste collectors). If you do not provide adequate protection and safety for yourself, you will lose your job and be in a poor state of health.”*
(Male IDI discussant from company A)

The study participants indicated that workplace uncertainties introduce health risks among waste collectors. The participants cited exposure to unexpected waste, waste separation processes, and a lack of knowledge among waste collectors of the degree of the health risk posed along the solid waste value chain as examples of workplace uncertainties that directly related to their work and health status. The participants’ views converged on the matter of unexpected waste as a major workplace uncertainty and health risk. The participants cited sharp objects, exposure to excreta/feces, and blood from animal carcasses, plastics, glass, paper cards, and diseased dead animals as examples of waste that they dealt with daily that posed a health risk to them, espoused one participant in a group discussion:


*“Hhhmm our health is indeed in our hands. We encounter different kinds of waste daily, from human to animal waste, solid and liquid. Most of these wastes are always at an advanced stage of decomposition. Depending on how well you are dressed (in PPE), you are always at risk… hmmm.”*
(Female FGD participant from company B)

The participants attributed the non-sorting of waste by waste generators during the process of disposal as one reason for their exposure to unexpected waste, as described by two participants:


*“Our waste is not sorted into various compartments. That is a major health threat in this job, where we encounter any kind and type of waste. So our health is really in our hands.”*
(Female IDI participant from company A)


*“Before you get to work, even if you are not provided with the right tools to protect yourself, you will have to get that to stay safe”*
(Female IDI participant from company B)

To address concerns over unexpected waste that posed a health risk and hazard to waste collectors, the participants suggested the need for strict legal regimes that prioritized waste sorting for individuals dumping waste and for waste companies in Ghana. The DWCs suggested that the number of years of work experience and exposure in the waste value chain also determined how knowledgeable they were. Furthermore, the participants described the degree of the health risk posed along the waste value chain, and how to reduce the health effects associated with their job, as narrated by one participant:


*“The company does not care about your health. I have been in this job for 7 years now and I know how to reduce the risk associated with it. My health is my concern. Otherwise, I would have left this job many years ago.”*
(Male IDI participant from company A)

When discussing the frequency and type of risk that heightened the health issues among the DWCs that compromised their health at work, the participants reported little concern about their health from employers. The DWCs indicated that they were not prioritized in any health schemes to support them and their dependents in the event of ill-health. The participants cited their employers’ inability even to provide a basic package of social health insurance to cover their health and medical needs yearly, given the potential risk and complaints of ill-health.

#### 3.1.2. Emergent Theme 2: Self-Reported Health Issues and Occupational Safety Coping Strategies among DWCs

Separate FGDs and IDIs with the participants revealed that constant back pain, flu, respiratory tract infections, wounds from sharp cuts from objects, anxiety/depression resulting in stress, and skin irritations were the commonest types of health problem cited among all the waste collectors. Injuries sustained from vehicle knockdowns were also cited as major health hazards associated with their work. The DWCs attributed their current state of health to their jobs, insisting that they had not received or sought any health care in relation to their level and type of health care need when they were not working as waste collectors, as expressed by FGD and IDI participants, respectively:


*‘’I have never known sickness or ill-health, my previous job as a gardener was less of a health burden, now I suffer from severe back pain because of the long hours under the sun picking and cleaning the markets.’’*
(Male FGD participant from company A)


*‘’Over the last 5 years, my health has worsened so much, a lot of the work requires manual and intensive labor here. I have never had a skin disease, but today, I visit the clinic almost every month because I constantly suffer from irritations on my skin…”*
(Female IDI participant from company B)

The dominant views around ill-health and diseases among waste collectors point to the fact that seeking health-care was a challenge for them, since they had no health insurance cover as domestic waste collectors, as remarked by one FGD participant:


*“We have no health insurance to meet any medical emergency that may arise from this work, so we don’t go to the clinic or hospital. Every day we have so many health issues that can happen when you are at work. So, we deal with it with the drugs we purchase from drug peddlers or the drug shops close by.”*
(Male FGD participant, Company B)

Thus, they coped by resorting to paying out-of-pocket for health care and treatment. Another coping strategy that they reported was that they engaged in self-medication, since they were not health-insured by their employers and they could not afford to pay their medical expenses. The common conditions that caused them to resort to self-medication were pain, headache, head, neck, or leg injuries, persistent coughs and flu, and wounds from severe bodily cuts from sharp objects. Pain relievers, antibiotics, sleeping pills, mixtures of local herbs (concoctions), and herbs were mentioned as some of the common medications to the participants used daily to deal with their health concerns. Although the DWCs admitted that self-medication or the use of poorly managed traditional treatment methods were not advisable practices, they still preferred to address their health issues themselves because of the cost of seeking health care in a health facility without health insurance.

Apart from direct injuries and disease conditions, the participants avowed that their psychosocial health and well-being were also affected daily because of work-related stress and other unmet psychosocial health needs. The respective IDIs and FGDs reported that stress and frequent burn-out due to the volume of workload daily were among the major factors that affected their physical and psychosocial health. They attributed the workload to many aspects of the manual labor that they performed. One focus group participant remarked that she felt the need to seek psychosocial support and counseling but did not know where to go because such important services were not provided for her at the workplace:


*“I feel stressed, and I am unable to perform my duties at an optimal level. Sometimes I find it very difficult to come to work, other times I feel like giving up on this job or to talk to someone about my problems but no one is near to listen.”*
(Female IDI discussant, company A)

Overall, the consequences of uncontrolled stress and heavy workloads resulted in psychological issues, including depression, lack of concentration at work, DWCs’ inability to form and maintain relationships at work, and reduced efficiency.

Both FGDs’ and IDIs’ interactions with waste collectors revealed that varied coping strategies were adopted to deal with occupational safety issues at the workplace. A dominant coping mechanism was the reliance on social support networks within and outside the workplace. The DWCs mentioned on several occasions how social networks (individuals and groups) provided both emotional and financial support to deal with adverse health situations, as well as social events, such as marriages and bereavement. Close social support networks, such as family and friends, often provide financial support to meet unexpected health needs, particularly in the case of accidents at work. Consensual views emerged on emotional management issues related to waste collectors’ ability to deal with anxiety/depression while still performing their assigned roles at the workplace. The participant stated that the frequent situations in which they needed to practice emotional management were commonly triggered by low remuneration and poor safety standards at work, as well as dealing with large volumes of work, which caused occasional stress and work-related-related burnout.

Another dominant coping mechanism adopted by the DWCs to deal with occupational safety needs at the workplace was resorting to being present at work at all times, even when suffering from ill-health. Given that failure to report at work may result in loss of jobs or underpaid remuneration, the waste collectors preferred to practice presenteeism. Although this approach is detrimental to their health, as admitted by the waste collectors themselves, presenteeism is seen as a mechanism to deal with increasing workloads, since the work was predominantly manual and laborious.

Dealing with workplace accidents with “*moving first-aid kits*” was commonly referred to as one-way waste collectors coped with daily accidents and injuries while collecting and disposing of waste. The waste collectors carried in their pockets small pallets of antibiotics, ointments for pain relief, and drugs for pain and flu when at work:


*“You see this bag; I have everything inside to deal with any hurt or accidents while am on the job. Ask anyone, they will show one or more first aid items they have in their pockets too…. it’s always safe and good to take your health on the streets into your own hands.”*
(Male IDI participant, company B)

The participants’ worldview informed this dominant view that their work environments were too risky and that daily accidents should be managed by themselves rather than traveling several kilometers to a health facility or by their employers. Although the participants admitted that this was self-medication, they prioritized this as an important activity that enabled them to perform their respective roles without unnecessary interruptions.

#### 3.1.3. Emergent Theme 3: Lack of Prioritization of Occupational Health and Safety Policies (OHSP) and Practices in the Workplace

The waste collectors interviewed affirmed that policies on workplace safety did exist, but they largely remained unimplemented. The DWCs said that although the waste company managers often took their external collaborators seriously by highlighting the good policies that exist in their companies, they the employees did not benefit from the implementation of such policies. The DWCs’ views showed that the managers/supervisors of the waste companies created the impression that the safety policies were being duly implemented for the direct benefit of the domestic waste collectors, which was in contrast to the reality. While affirming that work safety policies do exist, the consensual views among most of the participants ranked the prioritization of their health by these policies as low, as remarked by two participants in separate discussions:


*“Hmmm, the company has good policies for workers, we only hear it but do not benefit from it. These policies are only spoken of in good meetings outside the company. We hope that we too will be taken care of one day and we will benefit from the policies.”*
(Female FGD participant, company A)


*“For the policies benefitting our health, hmmm, we are yet to see that. For policies, we hear about it. We have been told, just that we are yet to feel it. I think they must re-prioritize and put our health needs first because it is currently low or non-existent in the workplace. Something needs to be done for us.”*
(Female IDI participant, company B)

Some of the study participants avowed the need for government to perform strong regulatory and monitoring role of the waste companies in the country to ensure the protection of the workers in these environments, as remarked by one FGD participant.


*“I use to work for a company in Tema before coming to work here, I think government officers used to come regularly to monitor and even sometimes interview us. So, the company was serious about the health of workers. Ever since I joined this waste company, I have not seen any government official coming to supervise what we do, not alone how workers’ welfare and health is. Government must do that to help us.”*
(Male FGD participant, company B)

With regard to how tailored training has supported waste collectors to address their occupational health and safety practices, the participants noted that they had been trained in the past on personal protective equipment (PPE) use and the procedures for reporting hazards and health and safety responses at the workplace among others. However, a major limitation of this training mentioned by the participants was how well they were implemented and coordinated for their fullest benefit. For example, the participants suggested that the PPE required regular supplies and change-overs, as it was never available after the waste collectors were first issued with it. This meant that the waste collectors spent extra money to purchase items such as gloves, boots, nose masks, and other working gear once the company’s issues were worn out. According to the DWCs, this contradicts the companies’ policies to ensure that their DWCs are safeguarded against workplace hazards.


*“You see the gloves and nose mask am wearing (pointing to her hands and nose), I purchased these myself for the work since I never got a replacement over the last 3 years. I had to purchase out-of-pocket for use”*
(Female FGD participants, company B)

Although the waste collectors demonstrated a clear understanding of the procedures for reporting the risks and hazards in their work environment, they revealed that they did not prioritize these reporting procedures. They explained that DWCs who follow and report health concerns do not receive any proper health care or welfare support when they follow the reporting procedures. 


*“Why worry and go through these procedures at the workplace, when all you get is paperwork and to sit in front of your bosses to narrate your incident over and over again for sympathy and not for assistance now or in the future to avert such an occurrence. So many of our colleagues just take care of themselves with their financial resources if they are injured in the process.”*
(Male IDI participant, company A)

The respective FGDs in the two companies also revealed a consensus on the low priority given by employers to communicating health and safety risk issues periodically to the waste collectors to inform how health and safety matters in the workplace were addressed on a routine basis. The absence of this periodic information created mistrust and negative perceptions among the waste collectors, who felt that only high-ranking employees benefited from health and safety workplace policies in the companies.


*“Sometimes management does not communicate periodically risk issues and accidents that occur in the workplace to guide our future work efforts for the fear of worker attrition in the company. But that has its negative effects since we are not guided by past happenings to learn from. There is a need for management to change this practice.”*
(Male FGD participant, company A)

The DWCs recounted that although their work related directly to risks and hazards, there was no health-and-safety manager at the local level in their companies responsible for planning specifically for work-based policies and schemes that benefited their welfare directly. Some insisted that the government must ensure strict compliance from waste companies to develop and build the capacity of health-and-safety managers in districts/municipalities where waste companies operate.


*“We don’t have a clinic or even a medical officer employed by the company who takes care of our health or even the daily hazards that we encounter while at work (pointing to a scar on his hand).”*
(Male FGD participant, company A)


*“When you go to the big hospitals where everyone attends for health-seeking, you spend the whole day in wait for care while your work suffers. You can even get back from that health facility and be sanctioned for not coming to work”*
(Female IDI participant, company B)

## 4. Discussion

Using a phenomenological qualitative research design, this study explored DWCs’ perceptions of occupational safety and self-reported health issues in a city in Ghana. The findings revealed that waste company authorities do not prioritize DWCs’ health and safety at the workplace, with complaints of injuries, stress, and frequent burn-out due to huge workloads as the major health concerns faced during work. Overall, the DWCs reiterated that they resorted to coping mechanisms to keep safe and avoid occupational health concerns. However, the DWCs avowed the need for occupational health safety policies (OHSP) to be prioritized and regulated both by the companies and by national regulatory authorities to ensure strict compliance to promote occupational safety and well-being among DWCs.

An important finding was the DWCs’ mistrust of their employers’ concern for their health and well-being. This perception resulted in DWCs’ low level of trust in employers initiating the appropriate steps to protect the health and well-being of waste collectors and other staff directly involved in the daily collection and processing of waste. This can demotivate and frustrate DWCs and lead to poor work output and high attrition, with workers leaving for other waste companies. This perceived neglect also creates an attitude where DWCs can abandon or refuse to work based on threats to their health and safety, since they are likely not to have any support from their employer. Similar perceptions are reported in other studies in India, Nigeria, and Ethiopia, where waste workers do not trust employers to address their welfare needs, including broader concerns about the working conditions and the environment and occupational morbidities associated with their work [9,28,30]. The findings confirm that there is a high level of mistrust between DWCs and waste-company employers due to a poor organizational cultural climate that leads to low motivation to work [38,39]. Waste-company employers need to see DWCs’ mistrust, which is due to the detrimental effects of their work on their well-being and quality of life, as a threat to the important role they play in keeping the environment clean. As a preventive measure, the causes of mistrust from DWCs and the lack of concern of waste-company employers need to be investigated to address them holistically. This will require waste-company employers and relevant stakeholders, such as local assemblies and DWCs, to engage in regular and constant dialogue to discuss the potential causes of the mistrust felt by DWCs and employers’ lack of concern for DWCs’ well-being and safety. Such efforts should include engaging in broad consultations and the commitment of employers and employees to resolving the issues. In addition, there is a need for waste-company employers to see DWCs as team members who should be treated with respect and recognition, even in times of sanctions.

In addition, this study revealed that DWCs encounter several occupational health hazards, such as physical ailments, cuts from sharp objects, needle-prick injuries, slipping and falls, musculoskeletal problems (back and wrist pains), and wounds (e.g., puncture and laceration wounds) in their line of work. The occupational health concerns reported in this study are documented in other published studies on waste collectors or individuals working in waste or sanitation environments [40,41,42]. Waste workers have a disproportionately high risk of suffering from occupational injuries (more than five times higher than other workforces) [12,43]. The reasons for DWCs’ health-hazard risk vary across several environments, depending on the knowledge and awareness of the potential hazards of the work environment and other supply-side factors under the control of employers and governments [44,45]. This study found that residents’ inability to sort waste into various compartments during disposal further aggravates the health risk concerns of DWCs. Waste segregation can minimize the potential health risks for DWCs [46,47], since unsorted waste may contain hazardous substances from various sources that can be detrimental to the health of DWCs [31,48,49].

The DWCs reported that they suffered from psychosocial disorders, work-related stress, and frequent burnout due to their heavy workload. This corroborates findings from other studies that reported that DWCs suffered from psychosocial risk factors, work-related stress, and job dissatisfaction [28,50]. The consequences of work-related stress and heavy workloads may lead to psychological health issues such as depression, a lack of concentration at work, an inability to form and maintain relationships at work, and poor lifestyle choices, such as the use of tobacco, drugs, and alcohol [51,52,53]. To improve the psychosocial health, work-related stress, and frequent burnout among DWCs, there is a need for waste-company employers to create a conducive work environment in line with the rights of DWCs as recommended by Sustainable Development Goals (SDG) 6.2 and 6.3.

Furthermore, this study found that DWCs scarcely prioritized the use of PPE for work, nor did they adequately protect themselves during work to reduce possible health risks. In other studies, this poor attitude regarding PPE use has also been reported among waste workers [54,55]. DWCs’ inability to understand the health implications of their work could be among the reasons for their nonuse of PPEs [8,56]. Although the usage of PPE cannot prevent exposure to various kinds of injury, its regular use by DWCs can minimize their chances of exposure to hazardous substances contained in solid waste [57]. These findings not only call for employers of waste companies to make sufficient PPE available to DWCs, but also for them to frequently conduct a refresher training course on the occupational safety and health hazards associated with solid-waste collection and handling.

Our finding suggests that employers failed to supervise the compliance of waste workers in the use of appropriate safety working gears based on the International Labor Organization (ILO) recommendations for work safety. The ILO recommends that waste workers should be given or receive health and safety training, health checks, and monitoring in their jobs at their workplaces [58,59]. Although Ghana’s Labor Law (Labor Act, 2003) recommends safe and effective PPE use by employees, this appeared not to be the case according to the views among the DWCs in this study. To make waste workers’ jobs safer, the ILO recommends providing protection against risks, suggesting the use of PPE, such as gloves, safety boots or wellington boots, and tools to sort waste. Our findings suggest the need for DWCs be given health and safety training, health checks, and monitoring in their jobs [58]. Waste-company employers need to monitor the safety and health of DWCs by designing and implementing occupational safety and health-hazard intervention programs for DWCs. In addition, there is a need for awareness raising and capacity building that would help promote DWCs’ identification of occupational safety and hazards associated with their work as quickly as possible before they become aggravated.

Another significant finding is that the DWCs reported the absence of guaranteed health insurance cover to meet their emergency medical needs. This created a work environment in which the DWCs preferred to practice presentism, by reporting to work but putting in little effort. This fueled the DWCs’ perception of their employers as having no regard for them as an important workforce. The absence of social support systems, particularly workers’ unions, to negotiate and engage with management over DWCs’ welfare needs reinforced the workers’ weak bargaining position. The lack of health insurance cover for persons involved in waste picking and disposal has been reported elsewhere [60]. In this study, we found that out-of-pocket payments, self-medication, and health-seeking delays were common coping strategies adopted due to unmet health insurance needs. To improve trust and guarantee better health outcomes and conditions for DWCs, waste-company employers need to take proactive measures by providing insurance cover for DWCs to help improve their health and well-being. In addition, interventions such as policies on the treatment of injured workers should be developed and implemented to help reduce the financial burden of medical care on DWCs. Creating a safe environment, in which DWCs feel that their concerns are shared by their employers, would improve DWCs’ motivation.

The findings in this study revealed that there is non-compliance with and non-enforcement of health and work safety policies to safeguard DWCs’ health and safety at the workplace. One of the reasons the DWCs viewed their companies as not prioritizing health and safety policy matters was the lack of regular health and safety training or capacity-building sessions for DWCs to identify and mitigate risk at work in accordance or compliance with national and international laws and standards. This situation was made worse due to the inability of public regulatory agencies to enforce laws and strictly monitor the compliance of waste companies with existing health and safety policies designed to safeguard employees [31,61]. To address these issues, DWCs should be involved in drafting solid waste-management policies. This would ensure that their interests and needs were factored into policy decision-making. Furthermore, there is a need for legislation requiring waste-company employers to conduct periodic medical examinations and screenings of their DWCs. Finally, at the national level, policies are required to ensure the full compliance of waste-company employers to ensure health and safety standards, particularly the use of PPE and effective measures for checking the occupational safety and health hazards experienced by DWCs.

### Study Limitations

This study has some limitations. The study was conducted on two waste companies operating in the study setting. Although the interviews attained saturation, the findings cannot be generalized because, as is typical in qualitative research, the participants were purposively sampled. Nevertheless, the findings highlight critical occupational health concerns that require both national and institutional changes to improve health work and safety concerns among DWCs in Ghana. Only the views of DWCs were sought regarding their occupational health and waste company policy matters. We acknowledge that a potential bias could have arisen, since the views of the employers were not sought in this study. However, to control the bias in the study, we chose study participants who were representative of the target population, and the results and findings were reviewed by the study participants and their peers. Any future study that triangulates views across multiple stakeholders on the occupational health concerns of waste collectors will provide additional knowledge and value in the context of Ghana or similar cultures.

## 5. Conclusions

This phenomenological qualitative research study aimed to examine domestic waste collectors’ perceptions of occupational safety and self-reported health outcomes in a city in Ghana. The study found that DWCs are exposed to multifaceted occupational safety and health hazards, such as injuries, diseases, musculoskeletal disorders such as back pain, and psychosocial disorders in their work. As a result, the study concluded that these factors should be carefully considered when developing and implementing occupational health and safety measures, as disregarding them could pose a potential risk to DWCs’ health and safety and well-being.

### 5.1. Policy Implications

There is a need for policies and regulatory measures that take into account interventions that promote social inclusion and occupational support for DWCs. This is due to the important roles played by DWCs in solid-waste collection and keeping the environment clean. Therefore, there is a need for recognition and adequate protection for their well-being and quality of life. There is also a need for social interventions and health insurance schemes that are geared towards DWCs and other vulnerable groups. Furthermore, there is a need for the adoption of capacity building and awareness creation that ensures the availability of insurance cover, as well as trust, in a system that addresses the currently unmet needs of DWCs. To improve occupational safety, health hazards, and well-being among DWCs, there is a need to design intervention programs that take into account the factors that expose DWCs to occupational safety and health hazards. Furthermore, workplace policies that target and provide psychosocial counseling and assistance for DWCs are needed to help them better manage work-related stress. In addition, policies are required at the local and national level to ensure full compliance by waste-company employers with occupational safety and health protocols, such as the use of PPE, to minimize occupational safety and health hazards and improve work-related stress and psychosocial disorders. In addition, policies on the treatment of injured workers should be developed and implemented to help reduce the financial burden of injury treatment on DWCs. DWCs should be included in the development of solid-waste management policies. In addition, there should be a policy and legislation requiring the managers of waste companies to conduct periodic medical examinations and screenings of their DWCs.

### 5.2. Recommendations

It is recommended that waste-company employers have risk-mitigation plans that consider the use of PPE and the design and execution of health and safety policies, to minimize DWCs’ exposure to occupational hazards and accidents. To ensure that avoidable accidents and health hazards are avoided, practical procedures and policies must be implemented. DWCs’ safety and health issues must be integrated into the organizational structure if DWCs’ safety is to be ensured.

The effectiveness of workplace health-and-safety management necessitates a strong commitment from waste-company employers as well as teamwork between safety and health specialists, DWCs and their associations/organizations, and management. Government agencies mandated with the responsibility for ensuring health and safety at the workplace need to be empowered to enforce occupational safety and health protocols and guidelines for accident prevention.

There is a need for further research to investigate the health and safety hazards associated with solid-waste collection in Ghana in order to guide the design and implementation of health-promotion measures to protect the health and safety of DWCs in Ghana. Furthermore, studies can be undertaken to assess the commitment of solid-waste-company employers and various district and municipal assemblies in Ghana to safeguarding the health and safety of DWCs. A future study that covers a wider geographic area is also recommended to unearth further contextual factors based on the objectives of this study.

## Figures and Tables

**Figure 1 ijerph-19-06539-f001:**
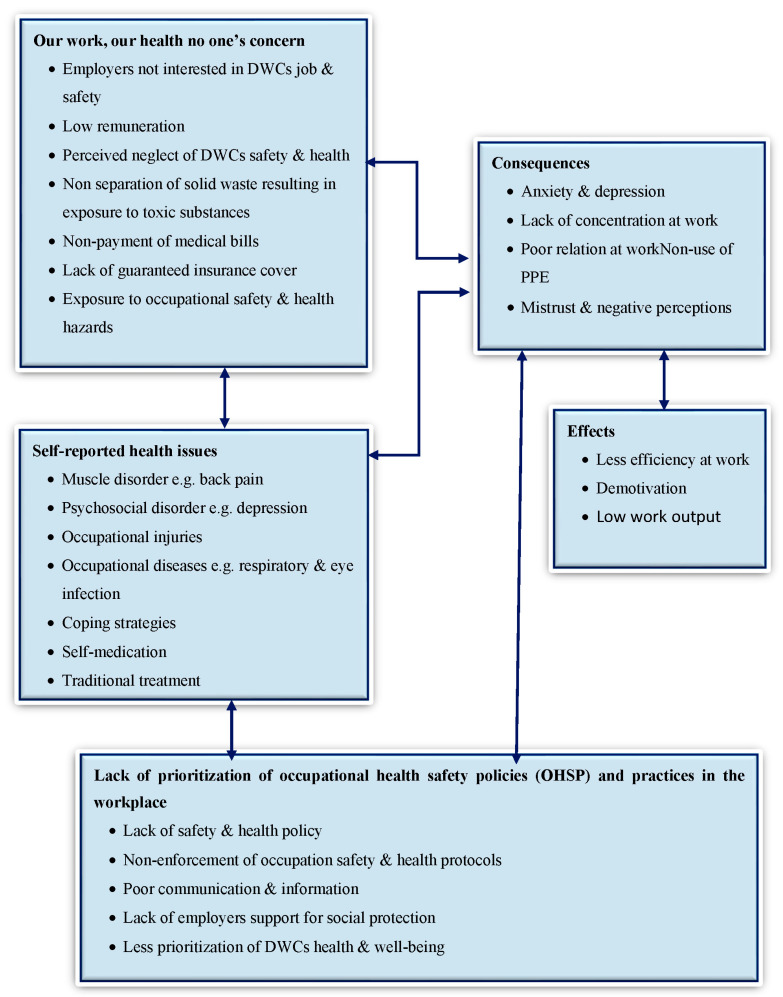
Presents a framework of factors in the study that influence DWCs’ perceptions of occupational safety and self-reported health outcomes.

**Table 1 ijerph-19-06539-t001:** Categories and number of qualitative interviews conducted.

Type of Interview	Participants (DWCs)	Number (n)
In-Depth Interviews	Company A	10
	Company B	5
	Total IDIs	15
Focus Group Discussions	Company A	33
	Company B	16
	Total FGDs	49

**Table 2 ijerph-19-06539-t002:** Interview guide questions in the FGDs and IDI.

1. Nature of hazards associated with waste collection in the study area *Can you indicate the nature of health hazards associated with your work? Probe further if the participant names any of the following.* - Cuts and pricks from sharp objectives such as needles, scrap metals, broken glasses, etc. - Harsh weather conditions (exposure to the scorching sun, wind, and rain) - Bites by rodents (such as mice and rats), insects - Infections transmitted by insects - Exposure to excreta/feces and blood - Type, frequency, and perceived causes of self-reported health-related issues among DWCs - Other perceived causes of self-reported health-related issues.
2. Factors that can expose domestic waste collectors to occupational hazards Indicate the factors which in your view can expose you to the risk and occupational and health hazards in your workplace. *Probe further if the participant names any of the following.* - Low level of literacy among DWCs - Negligence on the part of DWCs - Attitude of DWCs toward health and safety at the workplace, - Health and safety training, personal protective equipment use - Management support for DWCs - Knowledge about personal protective equipment use - Barriers to using personal protective equipment
3. Preventive measures for safety and health hazards at the workplace Do you know how/what health and safety measures are required in your workplace? *Probe further if the participant names any of the following.* - Formal health and safety policy - Hazards identification and assessment - Procedures for reporting hazards - Planned health and safety training for DWCs - Communicating health and safety performance to DWCs - Provision for first aid

**Table 3 ijerph-19-06539-t003:** Demographic and socio-economic characteristics of the study participants.

Characteristics	Company A		Company B	
	Manager/Supervisor	DWC	Manager/Supervisor	DWC
**Age in Years**	**n**	**n**	**n**	**n**
21–30	3	2	2	1
31–40	12	10	4	5
41–50	6	17	2	10
51–60	4	11	2	4
61+	-	3	-	1
**Ethnic Group of Participants**				
Ewe	22	42	10	19
Other	3	1	-	2
**Marital Status**				
Single	3	5	3	3
Married	22	27	7	13
Divorced/separated	-	3	-	1
Widow/widower	-	8	-	4
**Religion**				
Christian	18	27	8	14
Muslim	2	3	-	1
Traditionalist	2	11	1	5
No religion	3	2	1	1
**Educational Level**				
MSLC	-	11	-	5
JSS	-	9	-	5
Vocational training	-	3	-	2
None	-	20	-	9
SHS	15	-	8	-
Tertiary	10	-	2	-
**Gender**				
Female	4	29	2	13
Male	21	14	8	8
**Number of Years in the Workplace**				
Below 1 year	-	-	-	-
1–5 years	6	8	3	8
6–10 years	17	23	-	13
11–15 years	2	7	-	-
16–20 years	-	5	-	-
21 and above	-	-	-	-

MSLC—middle-school leaving certificate, JSS—junior secondary school, SHS—senior high school, n—number of study participants.

## Data Availability

The data generated, analyzed, and presented in this study are available from the authors upon request.

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
