# Peer review of "“Our Work, Our Health, No One’s Concern”: Domestic Waste Collectors’ Perceptions of Occupational Safety and Self-Reported Health Issues in an Urban Town in Ghana"

_ijerph, 2022, doi:10.3390/ijerph19116539_

Round 1

Reviewer 1 Report

This paper investigates the question on domestic waster collectors’ perceptions of occupational safety and self-reported health issue in an urban city in Ghana. There are 64 subjects interviewed to collect the data by using the convenience sampling method. And data are analyzed by processes of induction and deduction in NVivo that is a qualitive analysis software.  The authors conclude/find that subjects interviewed lost trust in their employers, because they believe their employers are not interested in their job safety and health. 

I believe the topic is significant in reporting the findings and potentially helpful to improve the health care of the population. But I have several comments and hope can enhance the paper.

  1. The authors used ‘convenience sampling’ method, which is non-probability sampling method and has many advantages, like collecting the data in a quick and low-cost way. But this approach may introduce bias. I would like to see the authors discuss this issue in the paper that how the bias is controlled when they make the inference.
  2. The scope is limited in a city in Ghana. If the study can expand the scope, say to the whole country, I think the significance of the paper will be improved.
  3. The results are partly demonstrated in the way of ‘testimony’. If possible, I would suggest that authors show the results summarized in a table or visualized by using a figure, which is more straightforward and readable for readers.
  4. There are many format issues need to be corrected:
    1. Line 14, font size.
    2. Line 73-75, font size.
    3. Line 81.
    4. Line 111, missing dot.
    5. Line128, extra dot.

Thanks.  

Author Response

Response to Reviewer 1 Comments

Thank you very much for your kind suggestions and comments.

Comments and Suggestions for Authors

This paper investigates the question on domestic waster collectors’ perceptions of occupational safety and self-reported health issue in an urban city in Ghana. There are 64 subjects interviewed to collect the data by using the convenience sampling method. And data are analyzed by processes of induction and deduction in NVivo that is a qualitive analysis software.  The authors conclude/find that subjects interviewed lost trust in their employers, because they believe their employers are not interested in their job safety and health. 

I believe the topic is significant in reporting the findings and potentially helpful to improve the health care of the population. But I have several comments and hope can enhance the paper.

Point 1: The authors used ‘convenience sampling’ method, which is non-probability sampling method and has many advantages, like collecting the data in a quick and low-cost way. But this approach may introduce bias. I would like to see the authors discuss this issue in the paper that how the bias is controlled when they make the inference.

Response 1: We thank the reviewer for raising this issue. In the revised manuscript, we explain the measures we took to control possible bias in the participants’ answers. Kindly see lines 162-176, and in the study, limitation lines 670-683.

Point 2: The scope is limited in a city in Ghana. If the study can expand the scope, say to the whole country, I think the significance of the paper will be improved.

Response 2: Thank you very much for the kind suggestion. Unfortunately, extending the scope to the whole nation is not within our capacities in terms of time, financial resources, and manpower. However, we see your point and have made reference to it in the recommendation section.

Point 3: The results are partly demonstrated in the way of ‘testimony’. If possible, I would suggest that the authors show the results summarized in a table or visualized by using a figure, which is more straightforward and readable for readers.

Response 3:  We have embraced the suggestion of the reviewer and have summarized our findings in a figure. See lines 219-279 for figure 1.)

Point 4: There are many format issues that need to be corrected:

    1. Line 14, font size.
    2. Line 73-75, font size.
    3. Line 81.
    4. Line 111, missing dot.
    5. Line128, extra dot.

Response 4: We thank the reviewer for pointing this out. We attended to the issues raised by the reviewer, and carefully went through the whole manuscript to fix textual and format issues.

Reviewer 2 Report

Dear authors.,

Congratulation!!!

Your study is quite interesting. However, there are some suggestions and comments to improve your quality manuscript. You can refer to the attached file.

Author Response

Response to Reviewer 2 Comments

Thank you very much for your kind commendation, suggestions, and comments.

Point 1: The author needs to add conclusions and policy implications to theory and practice here.

Response 2: We have now added conclusions and policy implications with regard to theory and practice to the abstract. See lines 36-39.

Point 2: The author needs to add research gaps, argue the previous studies, and focus on new knowledge (novelty) in the studies that previously were not studied.

Response 2: We have seen the reviewer’s point, and have added text to argue for the novelty of our research in the light of previous work done and omissions we see in the work. See lines 94-102.

Point 3: Study design and setting

Response 3: We thank the reviewer for the suggestion for the deletion of the word study. This has been deleted and it now reads as ‘’Research design and setting’’ in the manuscript. See line 110.

Point 4: Study Population

Response 4: We thank the reviewer for the recommendation. The study has been deleted from the study population and it now reads as ‘’Population’’ in the revised manuscript. See line 141.

Point 5: No need to put it in the BOX. Please change in Table Type or can in Paragraph.

Response 5: We thank the reviewer for the commendation for a table to be provided for a clearer understanding of the interview guide questions for the FGDs and IDI. This has been changed into a Table form as recommended. See lines 201- 203.

Point 6: You can provide the table for the study participants (demography profile)

Response 6: We thank the reviewer for the suggestion concerning a table to be provided for the study participants (demography profile). This has been provided and inserted in the revised manuscript.  See Table 3. in lines 297-299

Point 7: Change with “a phenomenological qualitative research design

Response 7: We thank the reviewer for the suggestion. This has been changed from a qualitative inquiry approach into "a phenomenological qualitative research design" in the revised manuscript. See line 547.

Point 8: The author requested to write policy implications separately to theory and practice.

Response 8: We thank the reviewer for the recommendation that policy implications be separated to theory and practice. This has been written separately in the revised manuscript. See lines 694-715.

Point 9: Also, include Limitations and Recommendations for Future Research

Response 9; We have seen the reviewer’s important suggestions. The study limitation and recommendations for future research have been explained and written separately in the revised manuscript.  See lines 670-83 and 717-737 respectively

Point 10: The author needs to focus on the research objectives and conclude it—no need to talk a lot about the research findings or some introduction again

Response 10; We agree with the reviewer about the need for the research to focus on the objectives and conclusion. The authors have addressed the issues to focus on the research objectives and conclusion in the revised manuscript. See lines 684-692.

Response to Reviewer 2 Comments

Thank you very much for your kind commendation, suggestions, and comments.

Point 1: The author needs to add conclusions and policy implications to theory and practice here.

Response 2: We have now added conclusions and policy implications with regard to theory and practice to the abstract. See lines 36-39.

Point 2: The author needs to add research gaps, argue the previous studies, and focus on new knowledge (novelty) in the studies that previously were not studied.

Response 2: We have seen the reviewer’s point, and have added text to argue for the novelty of our research in the light of previous work done and omissions we see in the work. See lines 94-102.

Point 3: Study design and setting

Response 3: We thank the reviewer for the suggestion for the deletion of the word study. This has been deleted and it now reads as ‘’Research design and setting’’ in the manuscript. See line 110.

Point 4: Study Population

Response 4: We thank the reviewer for the recommendation. The study has been deleted from the study population and it now reads as ‘’Population’’ in the revised manuscript. See line 141.

Point 5: No need to put it in the BOX. Please change in Table Type or can in Paragraph.

Response 5: We thank the reviewer for the commendation for a table to be provided for a clearer understanding of the interview guide questions for the FGDs and IDI. This has been changed into a Table form as recommended. See lines 201- 203.

Point 6: You can provide the table for the study participants (demography profile)

Response 6: We thank the reviewer for the suggestion concerning a table to be provided for the study participants (demography profile). This has been provided and inserted in the revised manuscript.  See Table 3. in lines 297-299

Point 7: Change with “a phenomenological qualitative research design

Response 7: We thank the reviewer for the suggestion. This has been changed from a qualitative inquiry approach into "a phenomenological qualitative research design" in the revised manuscript. See line 547.

Point 8: The author requested to write policy implications separately to theory and practice.

Response 8: We thank the reviewer for the recommendation that policy implications be separated to theory and practice. This has been written separately in the revised manuscript. See lines 694-715.

Point 9: Also, include Limitations and Recommendations for Future Research

Response 9; We have seen the reviewer’s important suggestions. The study limitation and recommendations for future research have been explained and written separately in the revised manuscript.  See lines 670-83 and 717-737 respectively

Point 10: The author needs to focus on the research objectives and conclude it—no need to talk a lot about the research findings or some introduction again

Response 10; We agree with the reviewer about the need for the research to focus on the objectives and conclusion. The authors have addressed the issues to focus on the research objectives and conclusion in the revised manuscript. See lines 684-692.

Response to Reviewer 2 Comments

Thank you very much for your kind commendation, suggestions, and comments.

Point 1: The author needs to add conclusions and policy implications to theory and practice here.

Response 2: We have now added conclusions and policy implications with regard to theory and practice to the abstract. See lines 36-39.

Point 2: The author needs to add research gaps, argue the previous studies, and focus on new knowledge (novelty) in the studies that previously were not studied.

Response 2: We have seen the reviewer’s point, and have added text to argue for the novelty of our research in the light of previous work done and omissions we see in the work. See lines 94-102.

Point 3: Study design and setting

Response 3: We thank the reviewer for the suggestion for the deletion of the word study. This has been deleted and it now reads as ‘’Research design and setting’’ in the manuscript. See line 110.

Point 4: Study Population

Response 4: We thank the reviewer for the recommendation. The study has been deleted from the study population and it now reads as ‘’Population’’ in the revised manuscript. See line 141.

Point 5: No need to put it in the BOX. Please change in Table Type or can in Paragraph.

Response 5: We thank the reviewer for the commendation for a table to be provided for a clearer understanding of the interview guide questions for the FGDs and IDI. This has been changed into a Table form as recommended. See lines 201- 203.

Point 6: You can provide the table for the study participants (demography profile)

Response 6: We thank the reviewer for the suggestion concerning a table to be provided for the study participants (demography profile). This has been provided and inserted in the revised manuscript.  See Table 3. in lines 297-299

Point 7: Change with “a phenomenological qualitative research design

Response 7: We thank the reviewer for the suggestion. This has been changed from a qualitative inquiry approach into "a phenomenological qualitative research design" in the revised manuscript. See line 547.

Point 8: The author requested to write policy implications separately to theory and practice.

Response 8: We thank the reviewer for the recommendation that policy implications be separated to theory and practice. This has been written separately in the revised manuscript. See lines 694-715.

Point 9: Also, include Limitations and Recommendations for Future Research

Response 9; We have seen the reviewer’s important suggestions. The study limitation and recommendations for future research have been explained and written separately in the revised manuscript.  See lines 670-83 and 717-737 respectively

Point 10: The author needs to focus on the research objectives and conclude it—no need to talk a lot about the research findings or some introduction again

Response 10; We agree with the reviewer about the need for the research to focus on the objectives and conclusion. The authors have addressed the issues to focus on the research objectives and conclusion in the revised manuscript. See lines 684-692.

Response to Reviewer 2 Comments

Thank you very much for your kind commendation, suggestions, and comments.

Point 1: The author needs to add conclusions and policy implications to theory and practice here.

Response 2: We have now added conclusions and policy implications with regard to theory and practice to the abstract. See lines 36-39.

Point 2: The author needs to add research gaps, argue the previous studies, and focus on new knowledge (novelty) in the studies that previously were not studied.

Response 2: We have seen the reviewer’s point, and have added text to argue for the novelty of our research in the light of previous work done and omissions we see in the work. See lines 94-102.

Point 3: Study design and setting

Response 3: We thank the reviewer for the suggestion for the deletion of the word study. This has been deleted and it now reads as ‘’Research design and setting’’ in the manuscript. See line 110.

Point 4: Study Population

Response 4: We thank the reviewer for the recommendation. The study has been deleted from the study population and it now reads as ‘’Population’’ in the revised manuscript. See line 141.

Point 5: No need to put it in the BOX. Please change in Table Type or can in Paragraph.

Response 5: We thank the reviewer for the commendation for a table to be provided for a clearer understanding of the interview guide questions for the FGDs and IDI. This has been changed into a Table form as recommended. See lines 201- 203.

Point 6: You can provide the table for the study participants (demography profile)

Response 6: We thank the reviewer for the suggestion concerning a table to be provided for the study participants (demography profile). This has been provided and inserted in the revised manuscript.  See Table 3. in lines 297-299

Point 7: Change with “a phenomenological qualitative research design

Response 7: We thank the reviewer for the suggestion. This has been changed from a qualitative inquiry approach into "a phenomenological qualitative research design" in the revised manuscript. See line 547.

Point 8: The author requested to write policy implications separately to theory and practice.

Response 8: We thank the reviewer for the recommendation that policy implications be separated to theory and practice. This has been written separately in the revised manuscript. See lines 694-715.

Point 9: Also, include Limitations and Recommendations for Future Research

Response 9; We have seen the reviewer’s important suggestions. The study limitation and recommendations for future research have been explained and written separately in the revised manuscript.  See lines 670-83 and 717-737 respectively

Point 10: The author needs to focus on the research objectives and conclude it—no need to talk a lot about the research findings or some introduction again

Response 10; We agree with the reviewer about the need for the research to focus on the objectives and conclusion. The authors have addressed the issues to focus on the research objectives and conclusion in the revised manuscript. See lines 684-692.

Response to Reviewer 2 Comments

Thank you very much for your kind commendation, suggestions, and comments.

Point 1: The author needs to add conclusions and policy implications to theory and practice here.

Response 2: We have now added conclusions and policy implications with regard to theory and practice to the abstract. See lines 36-39.

Point 2: The author needs to add research gaps, argue the previous studies, and focus on new knowledge (novelty) in the studies that previously were not studied.

Response 2: We have seen the reviewer’s point, and have added text to argue for the novelty of our research in the light of previous work done and omissions we see in the work. See lines 94-102.

Point 3: Study design and setting

Response 3: We thank the reviewer for the suggestion for the deletion of the word study. This has been deleted and it now reads as ‘’Research design and setting’’ in the manuscript. See line 110.

Point 4: Study Population

Response 4: We thank the reviewer for the recommendation. The study has been deleted from the study population and it now reads as ‘’Population’’ in the revised manuscript. See line 141.

Point 5: No need to put it in the BOX. Please change in Table Type or can in Paragraph.

Response 5: We thank the reviewer for the commendation for a table to be provided for a clearer understanding of the interview guide questions for the FGDs and IDI. This has been changed into a Table form as recommended. See lines 201- 203.

Point 6: You can provide the table for the study participants (demography profile)

Response 6: We thank the reviewer for the suggestion concerning a table to be provided for the study participants (demography profile). This has been provided and inserted in the revised manuscript.  See Table 3. in lines 297-299

Point 7: Change with “a phenomenological qualitative research design

Response 7: We thank the reviewer for the suggestion. This has been changed from a qualitative inquiry approach into "a phenomenological qualitative research design" in the revised manuscript. See line 547.

Point 8: The author requested to write policy implications separately to theory and practice.

Response 8: We thank the reviewer for the recommendation that policy implications be separated to theory and practice. This has been written separately in the revised manuscript. See lines 694-715.

Point 9: Also, include Limitations and Recommendations for Future Research

Response 9; We have seen the reviewer’s important suggestions. The study limitation and recommendations for future research have been explained and written separately in the revised manuscript.  See lines 670-83 and 717-737 respectively

Point 10: The author needs to focus on the research objectives and conclude it—no need to talk a lot about the research findings or some introduction again

Response 10; We agree with the reviewer about the need for the research to focus on the objectives and conclusion. The authors have addressed the issues to focus on the research objectives and conclusion in the revised manuscript. See lines 684-692.

Response to Reviewer 2 Comments

Thank you very much for your kind commendation, suggestions, and comments.

Point 1: The author needs to add conclusions and policy implications to theory and practice here.

Response 2: We have now added conclusions and policy implications with regard to theory and practice to the abstract. See lines 36-39.

Point 2: The author needs to add research gaps, argue the previous studies, and focus on new knowledge (novelty) in the studies that previously were not studied.

Response 2: We have seen the reviewer’s point, and have added text to argue for the novelty of our research in the light of previous work done and omissions we see in the work. See lines 94-102.

Point 3: Study design and setting

Response 3: We thank the reviewer for the suggestion for the deletion of the word study. This has been deleted and it now reads as ‘’Research design and setting’’ in the manuscript. See line 110.

Point 4: Study Population

Response 4: We thank the reviewer for the recommendation. The study has been deleted from the study population and it now reads as ‘’Population’’ in the revised manuscript. See line 141.

Point 5: No need to put it in the BOX. Please change in Table Type or can in Paragraph.

Response 5: We thank the reviewer for the commendation for a table to be provided for a clearer understanding of the interview guide questions for the FGDs and IDI. This has been changed into a Table form as recommended. See lines 201- 203.

Point 6: You can provide the table for the study participants (demography profile)

Response 6: We thank the reviewer for the suggestion concerning a table to be provided for the study participants (demography profile). This has been provided and inserted in the revised manuscript.  See Table 3. in lines 297-299

Point 7: Change with “a phenomenological qualitative research design

Response 7: We thank the reviewer for the suggestion. This has been changed from a qualitative inquiry approach into "a phenomenological qualitative research design" in the revised manuscript. See line 547.

Point 8: The author requested to write policy implications separately to theory and practice.

Response 8: We thank the reviewer for the recommendation that policy implications be separated to theory and practice. This has been written separately in the revised manuscript. See lines 694-715.

Point 9: Also, include Limitations and Recommendations for Future Research

Response 9; We have seen the reviewer’s important suggestions. The study limitation and recommendations for future research have been explained and written separately in the revised manuscript.  See lines 670-83 and 717-737 respectively

Point 10: The author needs to focus on the research objectives and conclude it—no need to talk a lot about the research findings or some introduction again

Response 10; We agree with the reviewer about the need for the research to focus on the objectives and conclusion. The authors have addressed the issues to focus on the research objectives and conclusion in the revised manuscript. See lines 684-692.

Response to Reviewer 2 Comments

Thank you very much for your kind commendation, suggestions, and comments.

Point 1: The author needs to add conclusions and policy implications to theory and practice here.

Response 2: We have now added conclusions and policy implications with regard to theory and practice to the abstract. See lines 36-39.

Point 2: The author needs to add research gaps, argue the previous studies, and focus on new knowledge (novelty) in the studies that previously were not studied.

Response 2: We have seen the reviewer’s point, and have added text to argue for the novelty of our research in the light of previous work done and omissions we see in the work. See lines 94-102.

Point 3: Study design and setting

Response 3: We thank the reviewer for the suggestion for the deletion of the word study. This has been deleted and it now reads as ‘’Research design and setting’’ in the manuscript. See line 110.

Point 4: Study Population

Response 4: We thank the reviewer for the recommendation. The study has been deleted from the study population and it now reads as ‘’Population’’ in the revised manuscript. See line 141.

Point 5: No need to put it in the BOX. Please change in Table Type or can in Paragraph.

Response 5: We thank the reviewer for the commendation for a table to be provided for a clearer understanding of the interview guide questions for the FGDs and IDI. This has been changed into a Table form as recommended. See lines 201- 203.

Point 6: You can provide the table for the study participants (demography profile)

Response 6: We thank the reviewer for the suggestion concerning a table to be provided for the study participants (demography profile). This has been provided and inserted in the revised manuscript.  See Table 3. in lines 297-299

Point 7: Change with “a phenomenological qualitative research design

Response 7: We thank the reviewer for the suggestion. This has been changed from a qualitative inquiry approach into "a phenomenological qualitative research design" in the revised manuscript. See line 547.

Point 8: The author requested to write policy implications separately to theory and practice.

Response 8: We thank the reviewer for the recommendation that policy implications be separated to theory and practice. This has been written separately in the revised manuscript. See lines 694-715.

Point 9: Also, include Limitations and Recommendations for Future Research

Response 9; We have seen the reviewer’s important suggestions. The study limitation and recommendations for future research have been explained and written separately in the revised manuscript.  See lines 670-83 and 717-737 respectively

Point 10: The author needs to focus on the research objectives and conclude it—no need to talk a lot about the research findings or some introduction again

Response 10; We agree with the reviewer about the need for the research to focus on the objectives and conclusion. The authors have addressed the issues to focus on the research objectives and conclusion in the revised manuscript. See lines 684-692.

Response to Reviewer 2 Comments

Thank you very much for your kind commendation, suggestions, and comments.

Point 1: The author needs to add conclusions and policy implications to theory and practice here.

Response 2: We have now added conclusions and policy implications with regard to theory and practice to the abstract. See lines 36-39.

Point 2: The author needs to add research gaps, argue the previous studies, and focus on new knowledge (novelty) in the studies that previously were not studied.

Response 2: We have seen the reviewer’s point, and have added text to argue for the novelty of our research in the light of previous work done and omissions we see in the work. See lines 94-102.

Point 3: Study design and setting

Response 3: We thank the reviewer for the suggestion for the deletion of the word study. This has been deleted and it now reads as ‘’Research design and setting’’ in the manuscript. See line 110.

Point 4: Study Population

Response 4: We thank the reviewer for the recommendation. The study has been deleted from the study population and it now reads as ‘’Population’’ in the revised manuscript. See line 141.

Point 5: No need to put it in the BOX. Please change in Table Type or can in Paragraph.

Response 5: We thank the reviewer for the commendation for a table to be provided for a clearer understanding of the interview guide questions for the FGDs and IDI. This has been changed into a Table form as recommended. See lines 201- 203.

Point 6: You can provide the table for the study participants (demography profile)

Response 6: We thank the reviewer for the suggestion concerning a table to be provided for the study participants (demography profile). This has been provided and inserted in the revised manuscript.  See Table 3. in lines 297-299

Point 7: Change with “a phenomenological qualitative research design

Response 7: We thank the reviewer for the suggestion. This has been changed from a qualitative inquiry approach into "a phenomenological qualitative research design" in the revised manuscript. See line 547.

Point 8: The author requested to write policy implications separately to theory and practice.

Response 8: We thank the reviewer for the recommendation that policy implications be separated to theory and practice. This has been written separately in the revised manuscript. See lines 694-715.

Point 9: Also, include Limitations and Recommendations for Future Research

Response 9; We have seen the reviewer’s important suggestions. The study limitation and recommendations for future research have been explained and written separately in the revised manuscript.  See lines 670-83 and 717-737 respectively

Point 10: The author needs to focus on the research objectives and conclude it—no need to talk a lot about the research findings or some introduction again

Response 10; We agree with the reviewer about the need for the research to focus on the objectives and conclusion. The authors have addressed the issues to focus on the research objectives and conclusion in the revised manuscript. See lines 684-692.

Response to Reviewer 2 Comments

Thank you very much for your kind commendation, suggestions, and comments.

Point 1: The author needs to add conclusions and policy implications to theory and practice here.

Response 2: We have now added conclusions and policy implications with regard to theory and practice to the abstract. See lines 36-39.

Point 2: The author needs to add research gaps, argue the previous studies, and focus on new knowledge (novelty) in the studies that previously were not studied.

Response 2: We have seen the reviewer’s point, and have added text to argue for the novelty of our research in the light of previous work done and omissions we see in the work. See lines 94-102.

Point 3: Study design and setting

Response 3: We thank the reviewer for the suggestion for the deletion of the word study. This has been deleted and it now reads as ‘’Research design and setting’’ in the manuscript. See line 110.

Point 4: Study Population

Response 4: We thank the reviewer for the recommendation. The study has been deleted from the study population and it now reads as ‘’Population’’ in the revised manuscript. See line 141.

Point 5: No need to put it in the BOX. Please change in Table Type or can in Paragraph.

Response 5: We thank the reviewer for the commendation for a table to be provided for a clearer understanding of the interview guide questions for the FGDs and IDI. This has been changed into a Table form as recommended. See lines 201- 203.

Point 6: You can provide the table for the study participants (demography profile)

Response 6: We thank the reviewer for the suggestion concerning a table to be provided for the study participants (demography profile). This has been provided and inserted in the revised manuscript.  See Table 3. in lines 297-299

Point 7: Change with “a phenomenological qualitative research design

Response 7: We thank the reviewer for the suggestion. This has been changed from a qualitative inquiry approach into "a phenomenological qualitative research design" in the revised manuscript. See line 547.

Point 8: The author requested to write policy implications separately to theory and practice.

Response 8: We thank the reviewer for the recommendation that policy implications be separated to theory and practice. This has been written separately in the revised manuscript. See lines 694-715.

Point 9: Also, include Limitations and Recommendations for Future Research

Response 9; We have seen the reviewer’s important suggestions. The study limitation and recommendations for future research have been explained and written separately in the revised manuscript.  See lines 670-83 and 717-737 respectively

Point 10: The author needs to focus on the research objectives and conclude it—no need to talk a lot about the research findings or some introduction again

Response 10; We agree with the reviewer about the need for the research to focus on the objectives and conclusion. The authors have addressed the issues to focus on the research objectives and conclusion in the revised manuscript. See lines 684-692.

Response to Reviewer 2 Comments

Thank you very much for your kind commendation, suggestions, and comments.

Point 1: The author needs to add conclusions and policy implications to theory and practice here.

Response 2: We have now added conclusions and policy implications with regard to theory and practice to the abstract. See lines 36-39.

Point 2: The author needs to add research gaps, argue the previous studies, and focus on new knowledge (novelty) in the studies that previously were not studied.

Response 2: We have seen the reviewer’s point, and have added text to argue for the novelty of our research in the light of previous work done and omissions we see in the work. See lines 94-102.

Point 3: Study design and setting

Response 3: We thank the reviewer for the suggestion for the deletion of the word study. This has been deleted and it now reads as ‘’Research design and setting’’ in the manuscript. See line 110.

Point 4: Study Population

Response 4: We thank the reviewer for the recommendation. The study has been deleted from the study population and it now reads as ‘’Population’’ in the revised manuscript. See line 141.

Point 5: No need to put it in the BOX. Please change in Table Type or can in Paragraph.

Response 5: We thank the reviewer for the commendation for a table to be provided for a clearer understanding of the interview guide questions for the FGDs and IDI. This has been changed into a Table form as recommended. See lines 201- 203.

Point 6: You can provide the table for the study participants (demography profile)

Response 6: We thank the reviewer for the suggestion concerning a table to be provided for the study participants (demography profile). This has been provided and inserted in the revised manuscript.  See Table 3. in lines 297-299

Point 7: Change with “a phenomenological qualitative research design

Response 7: We thank the reviewer for the suggestion. This has been changed from a qualitative inquiry approach into "a phenomenological qualitative research design" in the revised manuscript. See line 547.

Point 8: The author requested to write policy implications separately to theory and practice.

Response 8: We thank the reviewer for the recommendation that policy implications be separated to theory and practice. This has been written separately in the revised manuscript. See lines 694-715.

Point 9: Also, include Limitations and Recommendations for Future Research

Response 9; We have seen the reviewer’s important suggestions. The study limitation and recommendations for future research have been explained and written separately in the revised manuscript.  See lines 670-83 and 717-737 respectively

Point 10: The author needs to focus on the research objectives and conclude it—no need to talk a lot about the research findings or some introduction again

Response 10; We agree with the reviewer about the need for the research to focus on the objectives and conclusion. The authors have addressed the issues to focus on the research objectives and conclusion in the revised manuscript. See lines 684-692.

Response to Reviewer 2 Comments

Thank you very much for your kind commendation, suggestions, and comments.

Point 1: The author needs to add conclusions and policy implications to theory and practice here.

Response 2: We have now added conclusions and policy implications with regard to theory and practice to the abstract. See lines 36-39.

Point 2: The author needs to add research gaps, argue the previous studies, and focus on new knowledge (novelty) in the studies that previously were not studied.

Response 2: We have seen the reviewer’s point, and have added text to argue for the novelty of our research in the light of previous work done and omissions we see in the work. See lines 94-102.

Point 3: Study design and setting

Response 3: We thank the reviewer for the suggestion for the deletion of the word study. This has been deleted and it now reads as ‘’Research design and setting’’ in the manuscript. See line 110.

Point 4: Study Population

Response 4: We thank the reviewer for the recommendation. The study has been deleted from the study population and it now reads as ‘’Population’’ in the revised manuscript. See line 141.

Point 5: No need to put it in the BOX. Please change in Table Type or can in Paragraph.

Response 5: We thank the reviewer for the commendation for a table to be provided for a clearer understanding of the interview guide questions for the FGDs and IDI. This has been changed into a Table form as recommended. See lines 201- 203.

Point 6: You can provide the table for the study participants (demography profile)

Response 6: We thank the reviewer for the suggestion concerning a table to be provided for the study participants (demography profile). This has been provided and inserted in the revised manuscript.  See Table 3. in lines 297-299

Point 7: Change with “a phenomenological qualitative research design

Response 7: We thank the reviewer for the suggestion. This has been changed from a qualitative inquiry approach into "a phenomenological qualitative research design" in the revised manuscript. See line 547.

Point 8: The author requested to write policy implications separately to theory and practice.

Response 8: We thank the reviewer for the recommendation that policy implications be separated to theory and practice. This has been written separately in the revised manuscript. See lines 694-715.

Point 9: Also, include Limitations and Recommendations for Future Research

Response 9; We have seen the reviewer’s important suggestions. The study limitation and recommendations for future research have been explained and written separately in the revised manuscript.  See lines 670-83 and 717-737 respectively

Point 10: The author needs to focus on the research objectives and conclude it—no need to talk a lot about the research findings or some introduction again

Response 10; We agree with the reviewer about the need for the research to focus on the objectives and conclusion. The authors have addressed the issues to focus on the research objectives and conclusion in the revised manuscript. See lines 684-692.

Response to Reviewer 2 Comments

Thank you very much for your kind commendation, suggestions, and comments.

Point 1: The author needs to add conclusions and policy implications to theory and practice here.

Response 2: We have now added conclusions and policy implications with regard to theory and practice to the abstract. See lines 36-39.

Point 2: The author needs to add research gaps, argue the previous studies, and focus on new knowledge (novelty) in the studies that previously were not studied.

Response 2: We have seen the reviewer’s point, and have added text to argue for the novelty of our research in the light of previous work done and omissions we see in the work. See lines 94-102.

Point 3: Study design and setting

Response 3: We thank the reviewer for the suggestion for the deletion of the word study. This has been deleted and it now reads as ‘’Research design and setting’’ in the manuscript. See line 110.

Point 4: Study Population

Response 4: We thank the reviewer for the recommendation. The study has been deleted from the study population and it now reads as ‘’Population’’ in the revised manuscript. See line 141.

Point 5: No need to put it in the BOX. Please change in Table Type or can in Paragraph.

Response 5: We thank the reviewer for the commendation for a table to be provided for a clearer understanding of the interview guide questions for the FGDs and IDI. This has been changed into a Table form as recommended. See lines 201- 203.

Point 6: You can provide the table for the study participants (demography profile)

Response 6: We thank the reviewer for the suggestion concerning a table to be provided for the study participants (demography profile). This has been provided and inserted in the revised manuscript.  See Table 3. in lines 297-299

Point 7: Change with “a phenomenological qualitative research design

Response 7: We thank the reviewer for the suggestion. This has been changed from a qualitative inquiry approach into "a phenomenological qualitative research design" in the revised manuscript. See line 547.

Point 8: The author requested to write policy implications separately to theory and practice.

Response 8: We thank the reviewer for the recommendation that policy implications be separated to theory and practice. This has been written separately in the revised manuscript. See lines 694-715.

Point 9: Also, include Limitations and Recommendations for Future Research

Response 9; We have seen the reviewer’s important suggestions. The study limitation and recommendations for future research have been explained and written separately in the revised manuscript.  See lines 670-83 and 717-737 respectively

Point 10: The author needs to focus on the research objectives and conclude it—no need to talk a lot about the research findings or some introduction again

Response 10; We agree with the reviewer about the need for the research to focus on the objectives and conclusion. The authors have addressed the issues to focus on the research objectives and conclusion in the revised manuscript. See lines 684-692.

Reviewer 3 Report

This interesting qualitative study highlights several needs and problems in this specific group of workers. To improve this manuscript, the authors should make some changes:
1. In the introductory section, what was the reason (main objective) for this qualitative study? The situation of the workers seen by themselves? The idea is not clear in the text.
2. In the material and methods section, it would be good to describe a little how the work of these DWCs is, what the schedules are, the uniforms, etc. to be able to compare them with the same workers in other countries. What was the rationale for the "DWCs should have worked for a year or more in a manager or supervisor role" criterion? Why did the participants need to work on this job? What is the reason for the distribution? "15 DWC participated in IDI and 49 participated in FGD." The description of the study design needs to be improved. In order to replicate this study, more details are needed. In ethical considerations, a paragraph should be added to this study that followed “the guidelines described in the Declaration of Helsinki”.
3. In the results section, it is a good idea to show a map of words (concept) on the perception of the WDC. Why do the authors use two fonts to show what the workers talked about?
4. In the discussion section, line 431... The DWC won't quit their job because of financial need, though, will they? Would this situation justify that their employers do not change the state of health and well-being of their employees?... the authors should be a little more critical about the role of employers, about the existing but not carried out the policy on the regulation of the welfare of workers. To be for the WDC an amplifier of its demands.

Author Response

Response to Reviewer 3 Comments

Comments and Suggestions for Authors

Thank you very much for your kind commendation, suggestions, and comments.

Point 1: What was the reason (main aim) for this qualitative study? The situation of the workers?

Response 1: The main aim of our research was not so much about the situation of the workers, but to identify and understand DWCs ' perceptions of occupational safety, and self-reported health outcomes. We have now explained this more clearly in the revised manuscript. See lines 94 -102.

Point 2: It would be good to describe a little what the work of these DWCs is like, what the hours are, the uniforms, etc. to be able to compare them with the same workers in other countries

Response 2: We agree with the reviewer that the reader could need a better picture of what the work of DWCs looks like. We have described this now in more detail. See lines 145-158.

Point 3: What is the reason for this ditribution (15 DWCs participated in IDIs and 49 participated in FGDs)

Response 3: We thank the reviewer for this question. The reason for having more participants in the FGD’s than in the IDI is partly the nature of both methods, which have been addressed and shown in table 1., in the revised manuscript. See lines 177-188.

Point 4: Why do the authors use to types of letters?

Response 4: We thank the reviewer for bringing this to our attention. We have addressed the reviewer's concerns and have gone over the entire manuscript again to correct textual and formatting errors.

Round 2

Reviewer 1 Report

Looks good. No further comments. Thanks!